

# Detection and identification of *Xanthomonas* pathotypes associated with citrus diseases using comparative genomics and multiplex PCR

Natasha P. Fonseca[1], Érica B. Felestrino[1], Washington L. Caneschi[1], Angélica B. Sanchez[1], Isabella F. Cordeiro[1], Camila G.C. Lemes[1], Renata A.B. Assis[1], Flávia M.S. Carvalho[2], Jesus A. Ferro[2], Alessandro M. Varani[2], José Belasque[3], Joao C. Setubal[4], Guilherme P. Telles[5], Deiviston S. Aguena[6], Nalvo F. Almeida[6] and Leandro M. Moreira[1,7]

[1] Núcleo de Pesquisas em Ciências Biológicas, Universidade Federal de Ouro Preto, Ouro Preto, Minas Gerais, Brazil

[2] Departamento de Tecnologia, Faculdade de Ciências Agrárias e Veterinárias de Jaboticabal, Universidade Estadual Paulista-Unesp, Jaboticabal, São Paulo, Brazil

[3] Departamento de Fitopatologia e Nematologia, Escola Superior de Agricultura ''Luiz de Queiroz'', Piracicaba, São Paulo, Brazil

[4] Departamento de Bioquímica, Instituto de Química, Universidade de São Paulo, São Paulo, Brazil

[5] Instituto de Computação, Universidade Estadual de Campinas, Campinas, São Paulo, Brazil

[6] Faculdade de Computação, Universidade Federal de Mato Grosso do Sul, Campo Grande, Mato Grosso do Sul, Brazil

[7] Departamento de Ciências Biológicas, Universidade Federal de Ouro Preto, Ouro Preto, Minas Gerais, Brazil

Corresponding authors
Nalvo F. Almeida,
nalvo@facom.ufms.br
Leandro M. Moreira,
lmmorei@gmail.com,
lmmorei@ufop.edu.br

## ABSTRACT

**Background.** In *Citrus* cultures, three species of *Xanthomonas* are known to cause distinct diseases. *X. citri* subsp. *citri* patothype A, *X. fuscans* subsp. *aurantifolii* pathotypes B and C, and *X. alfalfae* subsp. *citrumelonis*, are the causative agents of cancrosis A, B, C, and citrus bacterial spots, respectively. Although these species exhibit different levels of virulence and aggressiveness, only limited alternatives are currently available for proper and early detection of these diseases in the fields. The present study aimed to develop a new molecular diagnostic method based on genomic sequences derived from the four species of *Xanthomonas*.

**Results.** Using comparative genomics approaches, primers were synthesized for the identification of the four causative agents of citrus diseases. These primers were validated for their specificity to their target DNA by both conventional and multiplex PCR. Upon evaluation, their sensitivity was found to be 0.02 ng/µl *in vitro* and $1.5 \times 10^4$ CFU ml$^{-1}$ in infected leaves. Additionally, none of the primers were able to generate amplicons in 19 other genomes of *Xanthomonas* not associated with *Citrus* and one species of *Xylella*, the causal agent of citrus variegated chlorosis (CVC). This denotes strong specificity of the primers for the different species of *Xanthomonas* investigated in this study.

**Conclusions.** We demonstrated that these markers can be used as potential candidates for performing *in vivo* molecular diagnosis exclusively for citrus-associated *Xanthomonas*. The bioinformatics pipeline developed in this study to design specific

genomic regions is capable of generating specific primers. It is freely available and can be utilized for any other model organism.

## INTRODUCTION

*Xanthomonas* is a genus of phytopathogenic Gram-negative bacteria that cause a variety of diseases in several agricultural commodities, including citrus (*Vauterin et al., 1995*). These diseases cause huge economic losses to the agricultural sector. Consequently, bacteria from this genus have been targeted as an essential model of study in different pathosystems; for instance, *X. oryzae* in *Oryza* spp. (rice), *X. campestris* in *Brassica* spp. (cabbage), *X. vesicatoria* in *Solanum* spp. (potato), and *X. citri* in *Citrus* spp. (orange) (*Ferreira-Tonin et al., 2012*; *Huang et al., 2017*; *Mansfield et al., 2012*; *Ryan et al., 2011*; *Vauterin et al., 1990*). Since *Xanthomonas* are quarantine pests in all these pathosystems and controlling them with agrochemicals is very expensive, early detection is of paramount importance for the control and management of the diseases caused by these bacteria.

The *Xanthomonas* and citrus pathosystem is composed mainly of the following three species: *X. citri* subsp. *citri* pathotype A (*XccA*), *X. fuscans* subsp. *aurantifolii* (*Xau*), and *X. alfalfae* subsp. *citrumelonis* (*Xacm*). These bacteria damage crops and consequently cause losses (*Schaad et al., 2005*; *Schaad et al., 2006*) (Table 1). Although the pathosystem as here defined contains only three species, control and eradication efforts are complicated by the genetic diversity of the pathotypes, with varying degrees of aggressiveness and virulence (*Graham et al., 2004*; *Schubert et al., 2001*). *XccA*, the causative agent of Asian citrus canker, is the most aggressive and virulent among the pathotypes. It infects all cultivated varieties of citrus hosts (*Da Silva et al., 2002*). In addition to the pathotype A, two other variants, $A^*$ (*XccA\**) and $A^W$ (*XccA$^W$*) have been reported in previous studies (*Sun et al., 2004*; *Vernière et al., 1998*). Although they are genetically close to type A, these variants are associated with a smaller set of compatible hosts (Table 1) (*Graham et al., 2004*; *Vernière et al., 1998*). *Xau* is represented by two pathotypes known as B (*XauB*) and C (*XauC*). Pathotype B causes false citrus canker or cancrosis B, a less aggressive disease than citrus canker that affects sour lemons (*Citrus limon*) severely (*Schubert et al., 2001*; *Goto, Takahashi & Messina, 1980*). Pathotype C is the cause of cancrosis C, which is characterized by induction of hypersensitivity reaction in its hosts as described in 'Galego' acid lime (*C. aurantifolia*) (*Schubert et al., 2001*) and 'Swingle' citrumelo (*Citrus paradisi* × *Poncirus trifoliata*) (*Jaciani et al., 2009*). *Xacm*, or pathotype E, is the causative agent of citrus bacterial spot. It is not classified as cancrosis since it does not cause typical corticosteroid lesions. However, citrus bacterial spot may be confused with cancrosis, and pathotype E is pathogenic in 'Swingle' citrumelo (*Gottwald et al., 1991*; *Gottwald et al., 1988*; *Graham & Gottwald, 1990*; *Schoulties et al., 1987*).
**Table 1  Relation of compatible host and causal agents.**

| Host | Pathogen | | | | | |
|---|---|---|---|---|---|---|
| | *XccA* | *XccA*[a] | *XccA*[w] | *XauB* | *XauC* | *Xacm* |
| *Citrus sinensis* | +++ | − | − | + | ± | + |
| *Citrus aurantium* | +++ | − | − | + | − | + |
| *Citrus limon* | ++ | − | − | ++ | + | − |
| *Citrus limonia* | +++ | − | − | − | + | + |
| *Citrus latifolia* | ++ | − | − | − | + | + |
| *Citrus aurantiifolia* | +++ | + | + | + | +++ | + |
| *Citrus macrophylla* | + | − | + | − | − | − |
| *Citrus paradisi* × *Poncirus trifoliata* | +++ | − | − | − | ++ | + |
| *Citrus reticulata* (Tangerina 'Ponkan') | + | − | − | − | + | − |
| *Citrus reticulata* (Tangerina 'Cravo') | ++ | − | − | − | + | + |
| *Citrus reshni* | + | − | − | − | + | − |
| *Citrus paradisi* | +++ | − | − | − | − | + |
| *Citrus maxima* | + | − | − | + | − | − |

Notes.

Source: *Gottwald et al. (1991)*, *Gottwald et al. (1988)*, *Graham & Gottwald (1990)*, *Jaciani et al. (2012)*, *Schoulties et al. (1987)*, *Schubert et al. (2001)*, *Sun et al. (2004)* and *Vernière et al. (1998)*.

[a] The results presented in this table are means of several reports of the disease in plantations and studies of pathogenicity tests (*Schubert et al., 2001*; *Sun et al., 2004*; *Vernière et al., 1998*; *Gottwald et al., 1991*; *Gottwald et al., 1988*; *Graham & Gottwald, 1990*; *Schoulties et al., 1987*; *Jaciani et al., 2012*).

+++ very pathogenic to host.

++ moderately pathogenic to host.

+ little pathogenic to the host.

Due to this considerable diversity in the characteristics of infection and disease spread, these bacteria are one of the main phytosanitary problems that affect and decimate Brazilian orange orchards. As they are cosmopolitan pathogens, they not only harm the economy of Brazil, a major citrus producer, but also the world (*Jaciani et al., 2012*). Since the management of the plants during cultivation and export are different for each infecting strain, correct diagnosis of the pathogen may reduce damages (*Graham et al., 2004*).

Molecular diagnosis is a cost-effective option for identifying pathogens with great genetic proximity. Molecular biology techniques have been extensively used for identifying various species of the genus *Xanthomonas* (*Adikini et al., 2011*; *Coletta-Filho et al., 2006*; *Mavrodieva, Levy & Gabriel, 2004*; *Munhoz et al., 2011*; *Ngoc et al., 2009*; *Park et al., 2006*; *Rigano et al., 2010*; *Trindade et al., 2007*; *Waite et al., 2016*; *Zhao et al., 2016*). The genetic diversity of the *Xanthomonas* pathotypes that infect citus makes their correct identification a challenging task. The low sensitivity and false negative cases of serological tests (*Vernière et al., 1998*; *Gottwald et al., 2009*), and false positive amplifications of the molecular tests, demonstrate the need for new detection tests that are specific and that discriminate the different variants of this class of plant pathogens (*Delcourt et al., 2013*). Thus, this study aimed to develop a molecular detection method from unique genomic sequences, identified by comparative genomics, which can distinguish different *Xanthomonas* pathotypes infecting *Citrus* spp. These sequences are intended to be used only for detecting *Xanthomonas* isolates obtained from *Citrus* spp.

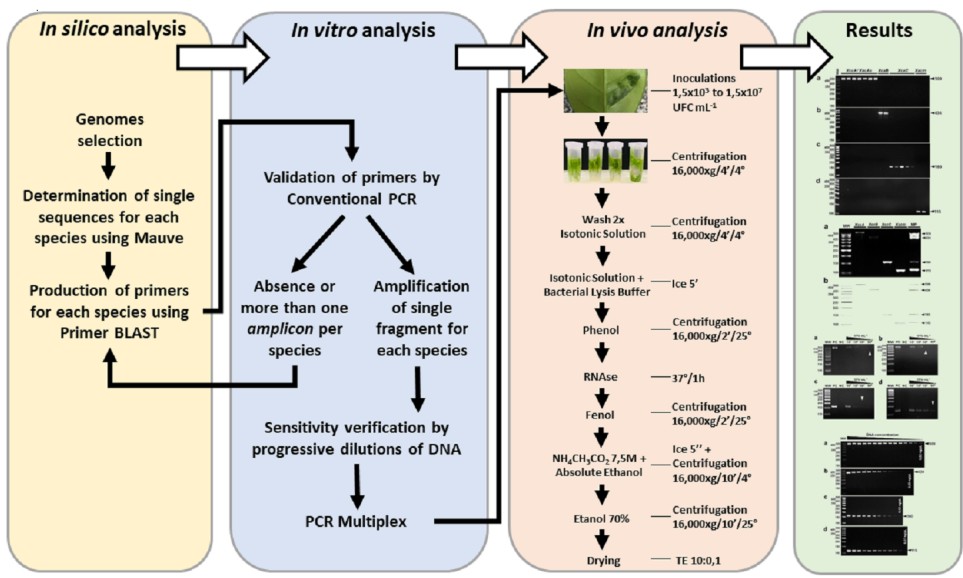

**Figure 1** **Experimental design.** Stages and procedures from the development of the *in silico* stages to the validation of the molecular results of the plant diagnosis.

## MATERIALS AND METHODS

For a better understanding of the methodological steps involved in this study, a flowchart of actions was established (Fig. 1).

### Comparative Genomics for identification of pathotype-specific regions

In order to identify the pathotype-specific regions, we have built a very simple 4-step pipeline. The input for the method is the set of nucleotide FASTA sequences from the four genomes of the genus *Xanthomonas,* belonging to each of the disease-causing-pathotypes in citrus obtained from NCBI (Table 2). The first step is to pre-process sequences, to concatenate all contigs of each genome, in case of being in a draft condition, and also to concatenate all FASTA genome files in a unique one. This file is the input for the second step, where mauveAligner, from the Mauve package (*Darling et al., 2004*), is used to build a multiple sequence alignment. MauveAligner outputs unique regions defined as regions that are present in one genome but absent from at least one of the other genomes. We call these unique regions 'Mauve islands', and they are the input to a custom Perl script to pick out specific regions from each genome. The third step is to post-process data just to separate all specific regions in distinct FASTA files. The candidate regions found at this point are double-checked in the fourth step, using BLASTn (*Altschul et al., 1997*). BLASTn is then performed on each candidate region from genome $G$ against the set of all genomes except $G$. The final output of this workflow consists of the list of all unique regions with no hits in this BLASTn run. This comparative genomics pipeline is freely available at https://git.facom.ufms.br/bioinfo/specific-primers, including a short and simple tutorial for installation and using, and can be used for any set of genomes in a regular Linux system.

**Table 2** General characteristics of the reference genomes.

| Organism | WDCM[a] (Accession number) | Abbreviations | Genome | NCBI reference sequence | CDSs | Ref |
|---|---|---|---|---|---|---|
| *X. citri* subsp. *citri* pathotype A strain 306 | IBSBF[b] (1594) | *XccA306* | Complete | NC_003919.1 | 4,687 | *Da Silva et al. (2002)* |
| *Xanthomonas citri* subsp. *aurantifolii* pathotype B strain 1561 | IBSBF (409) | *XauB1561* | Complete | CP011250.1 | 4,397 | AM Varani et al. (2017, unpublished data) |
| *Xanthomonas citri* subsp. *aurantifolii* pathotype C strain 1559 | IBSBF (381) | *XauC1559* | Complete | CP011160.1 | 4,733 | AM Varani et al. (2017, unpublished data) |
| *X. alfalfae* subsp. *citrumelonis* strain 1510 | IBSBF (1925) | *Xacm1510* | Draft | GCF_005059785.1[c] | 4,017 | AM Varani et al. (2019, unpublished data) |

**Notes.**
[a]WDCM (World Data Centre for Microorganisms).
[b]IBSBF—Instituto Biológico de São Paulo.
[c]*Xanthomonas euvesicatoria* pv. *citrumelonis* strain 1,510 is the same as *X. alfalfae* subsp. *citrumelonis* strain 1,510 (https://www.ncbi.nlm.nih.gov/nuccore/NZ_LAUO00000000.1).

Other specific regions finders are available in the literature, for instance Insignia (*Phillippy et al., 2009*). Insignia is a web application where the user can find specific regions from a genomic sequence, but only against sequences available currently in the previously populated database by the authors. On the other hand, our tool is a very easy-to-install and easy-to-use stand-alone tool where the user can find specific regions against any set of genomes (available in databases or not), even draft ones.

## Selection of specific sequences and production of primers

The sets of unique regions determined for each genome were used as a reference for the selection of the primers using Primer-BLAST tool (*Ye et al., 2012*). To verify the specificity of each primer to their target DNA, each generated primer was subjected to local alignment against *Xanthomonas* pathotypes that infect citrus using BLASTn. Primers that demonstrated specificity and ability to form amplicons of different sizes among the analyzed pathotype were selected for *in vitro* empirical verification.

## *In silico* evaluation of possible false positive amplifications

To verify false positive amplifications from other *Xanthomonas* species or other plant-associated microorganisms whose DNA could be extracted from infected plant tissue, the primers were compared against all sequences available at GenBank (using BLASTn against NT database).

## Bacterial strain and growth conditions

For the biological assays, 16 strains belonging to the pathotype *XccA* (7 strains), *XauB* (2 strains), *XauC* (5 strains), and *Xacm* (2 strains) were selected. These strains came from culture collections maintained by the Agronomy School of the University of São Paulo (ESALQ) and by the Faculty of Agrarian and Veterinary Sciences of the State University of São Paulo (Campus Jaboticabal), both in the state of São Paulo, Brazil. The isolates were

reactivated in nutrient agar medium—NA (5 g/l peptone, 3 g/l beef extract, 15 g/l agar, pH 6.8–7.0) and incubated at 28 °C for 48 h for *XccA* and *Xacm* strains and 96 h for *XauB* and *XauC* strains. Then, a colony was grown in nutrient broth medium (NB; NA without agar) under agitation at 250 rpm at 28 °C for 12 h.

In addition to the citrus-pathogenic *Xanthomonas* isolates, the markers have also been tested against other *Xanthomonas* strains that are not known to infect citrus: *X. axonopodis* pv. *vignicola* FDC 1701 (IBSBF 1739), *X. axonopodis* pv. *vesicatoria* 1387 (IBSBF 1387), *X. axonopodis* pv. *vesicatoria* FDC 1689 (IBSBF 331), *X. axonopodis* pv. *allii* 1770 (IBSBF 1770), *X. axonopodis* pv. *phaseoli* IAC 11280, *X. alfafae* subsp. *alfalfae* FDC 1711 (IBSBF 3174), *X. axonopodis* pv. *passiflorae* M89, *X. axonopodis* pv. *patelli* FDC 1727 (IBSBF 3181), *X. axonopodis* pv. *coracanae* FDC 1729 (IBSBF 3187), *X. axonopodis* pv. *manihotis* 1954 (IBSBF 1954), *X. axonopodis* pv. *phaseoli* var. *fuscans* IAC13755, *X. citri* pv. *rhynchosiae* FDC 1721 (IBSBF 3184), *X. axonopodis vasculorum* 2008 (IBSBF 2008), *X. axonopodis* pv. *axonopodis* FDC 1696 (IBSBF 1444), *X. axonopodis* pv. *dieffenbachiae* 1478 (IBSBF 1478), *X. axonopodis* pv. *glycines* FDC 1694 (IBSBF 333), *X. citri* pv. *cajani* FDC 1718 (IBSBF 3180), *X. citri* pv. *sesbaniae* FDC 1719 (IBSBF 3179), and against *Xylella fastidiosa* strain 9a5c (*Simpson et al., 2000*), a phytopathogenic bacterium that is disseminated by insects among citrus.

## Extraction of DNA from isolates

The DNA of the isolates was extracted with the *Spin Tissue*$^{TM}$ *Core Kit* (Qiagen Biosciences, MD, USA) following the manufacturer's recommendations.

## Inoculation of the isolates in the leaves and genomic DNA extraction from bacterial cells in exudate

The bacteria were inoculated into Hamlin-type orange leaves with a needleless syringe. The syringe was pressed lightly into the abaxial part of the leaf to allow entry of the phytopathogen into the intercellular space of the leaf. A leaf from the same plant was inoculated likewise with ultrapure water as a negative control. Immediately after the inoculation, the inoculated area was cut into small pieces, and bacterial exudation was carried out.

To mimic the condition of naturally infected plants, spray inoculation methods were also used. Spray inoculation was perform in 'Pera-rio' orange leaves according to the methods described by Yan and Wang (*Yan & Wang, 2011*). After 30 days of inoculation, leaves containing lesions were detached from the plants and each lesion was cut into small pieces to get the exudate with bacteria. The inoculated area/lesions were submerged in 1 mL of ultrapure water for 30 min and agitated for 5 min to facilitate bacterial exudation from the leaf. After agitation, genomic DNA extraction from bacterial cells in the exudate was performed with a rapid manual DNA extraction protocol. The exudate was centrifuged at 16,000 × $g$ for 4 min at 4 °C and washed twice with an isotonic solution (33 mM $KH_2PO_4$, 60 mM $K_2HPO_4$, 7.6 mM $(NH_4)_2SO_4$, 3.3 mM sodium citrate, and 4.0 mM $MgSO_4$). The pellet was resuspended in lysis buffer (10 mM Tris–HCl 1 M, pH 8.0; 100 mM EDTA 0.5 M, pH 8.0; 10 mM NaCl 5 M; 0.5% of 10% SDS), and isotonic solution with subsequent

extraction with phenol and incubation with RNAse (5 mg/ml) for 1 h at 37 °C. After further extraction with phenol, the DNA was precipitated with ethanol and sodium acetate (7.5 M, pH 7.4) and then diluted in Tris–EDTA buffer 10:0.1.

## PCR conditions

PCR amplifications were carried out in a final volume of 25µl. The final concentrations of the reagents in the reaction were: 2 mM $MgCl_2$, 200 µM dNTPs, 0.2 µM of each primer (Foward and Reverse), 1 unit of Taq DNA polymerase[TM] (*CellCo, SP, Brazil*) and 0.4 ng of the DNA sample from the isolates grown in medium or 1 µl of the total DNA extracted from the exudates. The initial denaturation temperature was 94 °C for 3 min, followed by 30 cycles of denaturation at 94 °C for 45 s, annealing at 59 °C for 45 s and extension at 72 °C for 45 s, followed by a final extension at 72 °C for 5 min. Verification of PCR amplicons was established by applying 10 µl of the PCR product in 3.0% (w/v) agarose gel.

## Multiplex PCR

Multiplex PCR amplifications (mPCR) were also performed in final volumes of 25 µl. The final concentration of the reagents in the reaction was the same as on the conventional PCR, except for increasing the concentration of dNTPs to 400 µM and using 0.2 µM of each primer (Forward and Reverse) of all four primer pairs (XccAm, XauBm, XauCm, and Xacmm) in the same reaction.

## Sensitivity of PCR

To verify the sensitivity of the PCR reaction *in vitro*, progressive dilutions of DNA from the isolates were performed until the amplification checked the minimal DNA concentration detectable by PCR. The isolates used in this test were *XccA* strain 306 (*XccA* 306), *XauB* strain FDC 1561 (*XauB* 1561), *XauC* strain FDC 1559 (*XauC* 1559), and *Xacm* strain 1510 (*Xacm* 1510). For leaf testing, sensitivity was found from minimal inoculations of leaf bacteria ($1.5 \times 10^7$, $1.5 \times 10^6$, $1.5 \times 10^5$, $1.5 \times 10^4$, $1.5 \times 10^3$ CFU ml$^{-1}$) with consequent lower extraction of DNA from the exudate.

## *In-silico* PCR

The *in-silico* PCR was performed using the tool '*In silico* PCR amplification' (*San Millán et al., 2013*) available at http://insilico.ehu.eus/.

# RESULTS

## Specific sequences and selection of primers

Mauve found 5,217 islands. From these, 173 candidates for specific regions of *XccA* 306, 59 of *XauB* 1561, 55 of *XauC* 1559 and 38 of *Xacm* 1510 were identified. After the BLAST refinement step, we found 135 specific regions of *XccA* 306, 23 of *XauB* 1561, 12 of *XauC* 1559, and 33 of *Xacm* 1510. From the sequences, a pair of primers were developed for each *Xanthomonas* pathotype. Each marker was developed using a different amplicon size, allowing its differentiation from the other pathotypes. The primer sequences and their respective amplicons are described in Table 3. Primer XccAm appears to amplify part of a hypothetical gene and a transcriptional activator FtrA, generating a 509bp amplicon.

**Table 3** **Molecular markers selected for the assays.** The oligonucleotides and amplicons generated by PCR are protected by the patent application registration under process number BR 10 2018 067332 7.

| Pathotypes | Markers | Forward | Reverse | Amplicon size (bp) | Tm (°C) |
|---|---|---|---|---|---|
| *XccA* | XccAm | ATGCTGAGCAAGCCTTCGAT | AGCTGGGAACGATGATGGTG | 509 | 59 |
| *XauB* | XauBm | TCGATCGCACGGACTACTTG | AAAATGCGGCTCTCCCTCTC | 434 | 59 |
| *XauC* | XauCm | CACTGGAGGCAGGAGTCGAG | CCACCCTCAAGTTCAGCAACA | 160 | 59 |
| *Xacm* | Xacmm | ACCAACACCTTGTGGTCGTA | TGTTCGTCAAACCGGCCA | 115 | 59 |

Primer XauBm amplifies an intergenic region (434 bp); primer XauCm amplifies part of a hypothetical gene (160 bp) and primer Xacmm amplifies part of a gene encoding a D-alanyl-D-alanine synthetase A (115 bp). Due to the intraspecific genetic proximity between *Xanthomonas*, a reference genome was used for each target pathotype.

BLASTn analysis of the primers against NCBI's non-redundant sequence database NT revealed high sequence specificity (100% identity) for the citrus-associated *Xanthomonas* pathothypes in all cases, and also for other non-citrus-associated *Xanthomonas* species (80–100% identity). However, *in silico* analysis of the possible amplicons generated in the non-citrus-associated species indicated different sizes from those expected for the defined targets. Only one pair of primers (*XauC*) indicated possible amplification in the case of *Stenotrophomonas maltophilia*, a bacterium that lives in aqueous environments, soil, and plants. However, the coverage of the XauCm forward and reverse primers on the genome of *Stenotrophomonas maltophilia* is not complete and the size of *in silico* amplicon is not similar to that of previously predicted amplicon for *XauC*.

## Specificity of molecular markers and sensitivity of PCR

The specificity for citrus-associated *Xanthomonas* verified *in silico* was confirmed as shown in Fig. 2. Primer XccAm generated an amplicon of size 509 bp, primer XauBm generated an amplicon of 434 bp, primer XauCm generated an amplicon of 160 bp, and primer Xacmm generated an amplicon of 115 bp for their corresponding pathotypes. All markers generated unique amplicons for their respective target DNAs without cross-amplifications for other investigated strains. Progressive dilutions of the target DNA verified the *in vitro* sensitivity. The minimum concentrations that allowed PCR amplification were 0.02 ng/μl for *XccA* (∼70 cells), 0.03 ng/μl for *XauB* (∼100 cells), 0.12 ng/μl for *XauC* (∼400 cells), and 0.07 ng/μl *Xacm* (∼200 cells) (Fig. 3).

These results were reproducible when the markers were subject to multiplex PCR with their respective target DNAs and with the DNAs of the four pathotypes (Fig. 4). Despite the findings *in silico*, when tested with the other 18 species of *Xanthomonas* that do not infect citrus, the markers developed for *XccA*, *XauB*, and *XauC* did not demonstrate amplification with any of these isolates (Fig. 5). In contrast, Xacmm amplified a weak fragment for five of the investigated strains (*X. axonopodis* pv. *vesicatoria* 1689, *X. axonopodis* pv. *passiflorae* M89, *X. axonopodis* pv. *manihotis* IBSBF1954, *X. axonopodis* pv. *axonopodis* 1696, and *X. axonopodis* pv. *glycines* 1694), but with a different amplicon size (∼125 bp) without correspondence to the established standards for citrus-associated xanthomonads. Although now empirically shown, this result does not affect detection since none of these strains

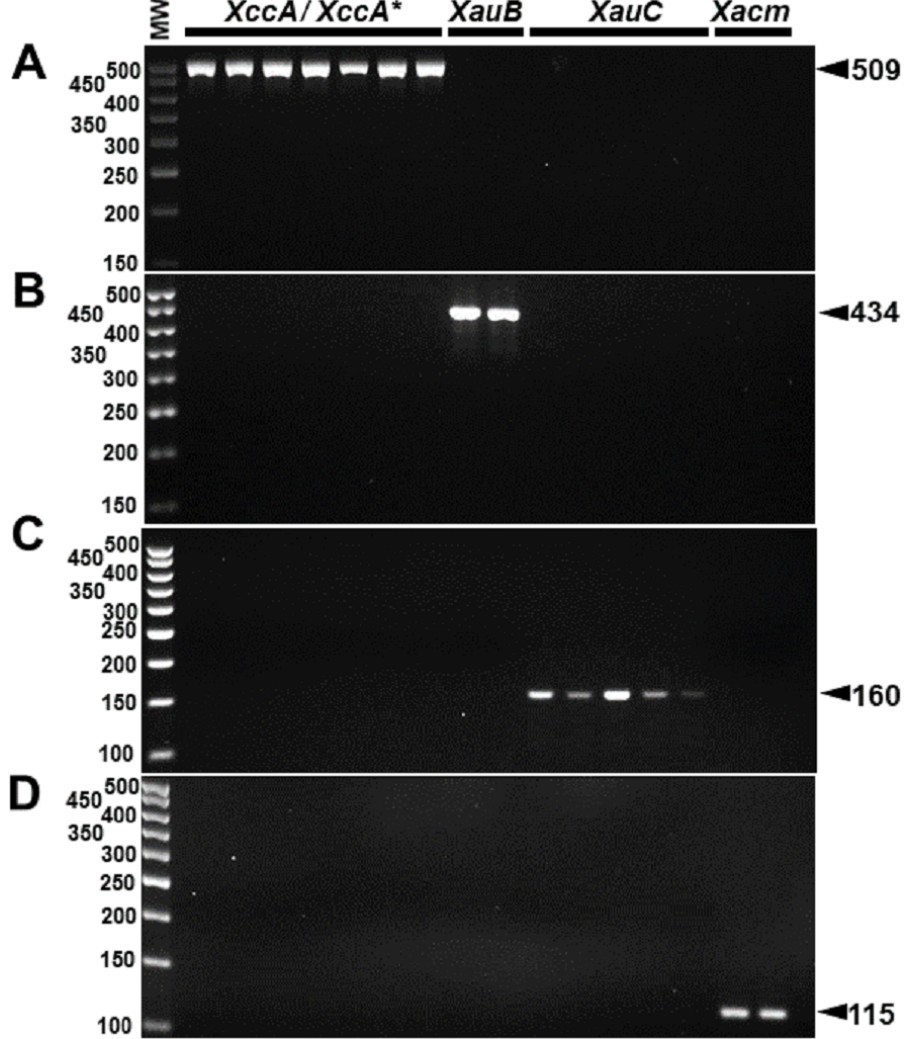

**Figure 2  Specificity of molecular markers *in vitro*.** Conventional PCR electrophoresis using Xcc-specific oligonucleotides for *Xanthomonas citri* subsp. *citri* pathotypes A (*XacA*) and A* (*XacA**) with a 509 bp amplicon, (A) *Xanthomonas fuscans* subsp. *aurantifolii* pathotype B (*XauB*) with 434 bp amplicon, (B) *Xanthomonas fuscans* subsp. *aurantifolii* pathotype C (*XauC*) with 160 bp amplicon, (C) *Xanthomonas alfalfae* subsp. *citrumelonis* (*Xacm*) with an amplicon of 115 bp, and (D) MW, molecular weight (ladder 100 bp).

with false positive amplification infects citrus. Finally, no amplicons were generated due to cross-amplification with DNA from *Xylella*, a plant pathogen that also infects citrus.

*In vivo* assays replicated the results obtained *in vitro*. The markers were specific for the inoculated bacterium (Fig. 6) and citrus canker lesions (Fig. 7) without cross-reaction with any other DNA that may have been extracted from exudate, including endophytic DNA. Additionally, the plants inoculated with water did not amplify with any marker tested, confirming the specificity for bacteria of the genus *Xanthomonas* (Fig. 6). For the exudate, the minimum concentrations that allowed PCR amplification were approximately 1.5 ×

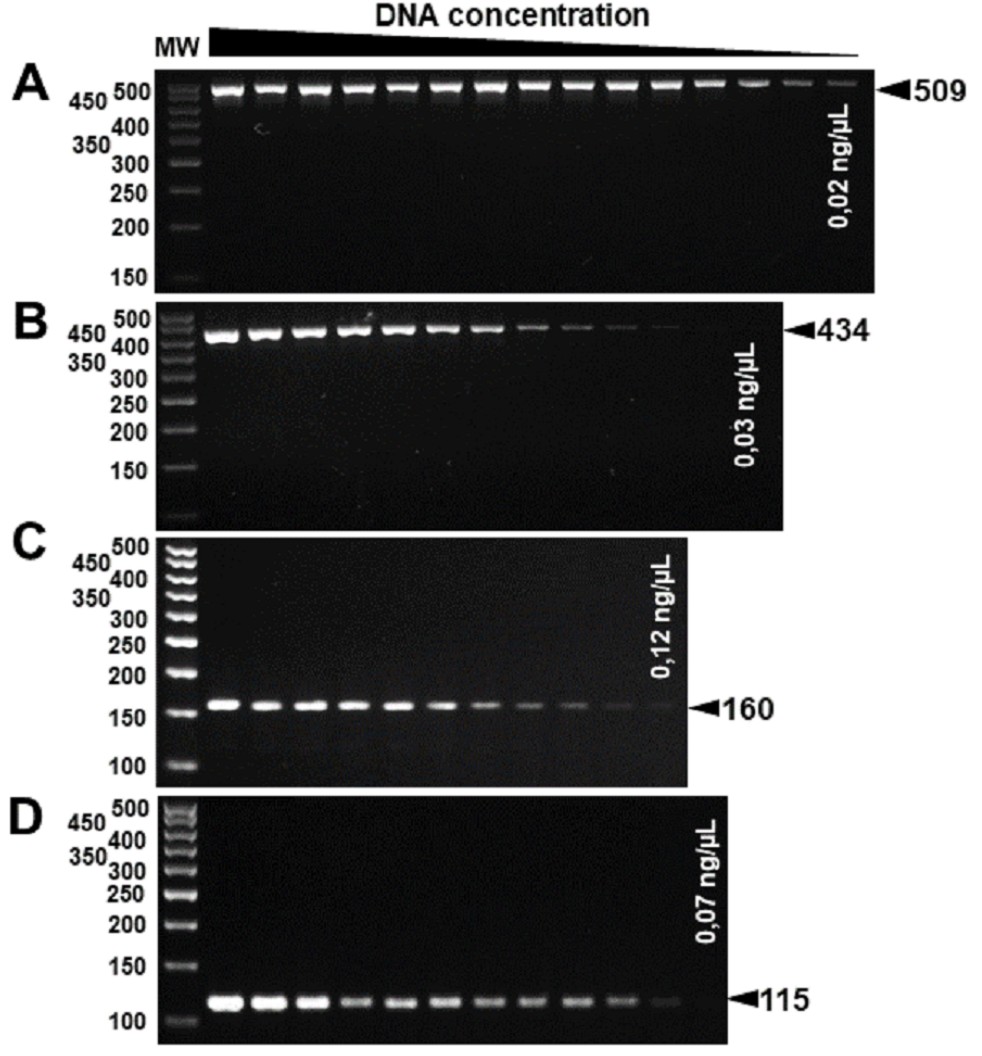

**Figure 3 Progressive DNA dilutions demonstrating the sensitivity of PCR.** (A) *XccA* (0.02 ng/μl, ~70 cells), (B) *XauB* (0.03 ng/μl, ~100 cells), (C) *XauC* (0.12 ng/μl, ~400 cells), and (D) *Xacm* (0.07 ng/μl, ~200 cells). MW, molecular weight (ladder 100 bp).

$10^4$ CFU mL$^{-1}$ for *XccA*, $1.5 \times 10^5$ CFU mL$^{-1}$ for *XauB*, $1.5 \times 10^5$ CFU mL$^{-1}$ for *XauC*, and $1.5 \times 10^4$ CFU mL$^{-1}$ for *Xacm* (Fig. 6).

## DISCUSSION

Currently, the citrus disease of greatest concern to citrus growers worldwide is greening, also known as Huanglongbin (HLB) (*Gottwald, Graça & Bassanezi, 2007*). However, citrus canker still is a problem for the citrus industry due to its virulence and rapid propagation characteristics (*Ference et al., 2017*). Coupled with deficiencies in maintaining legislation and inspecting contaminated orchards, citrus canker causes great losses to farmers worldwide (*Behlau, Fonseca & Belasque, 2016*). Thus, the correct identification of

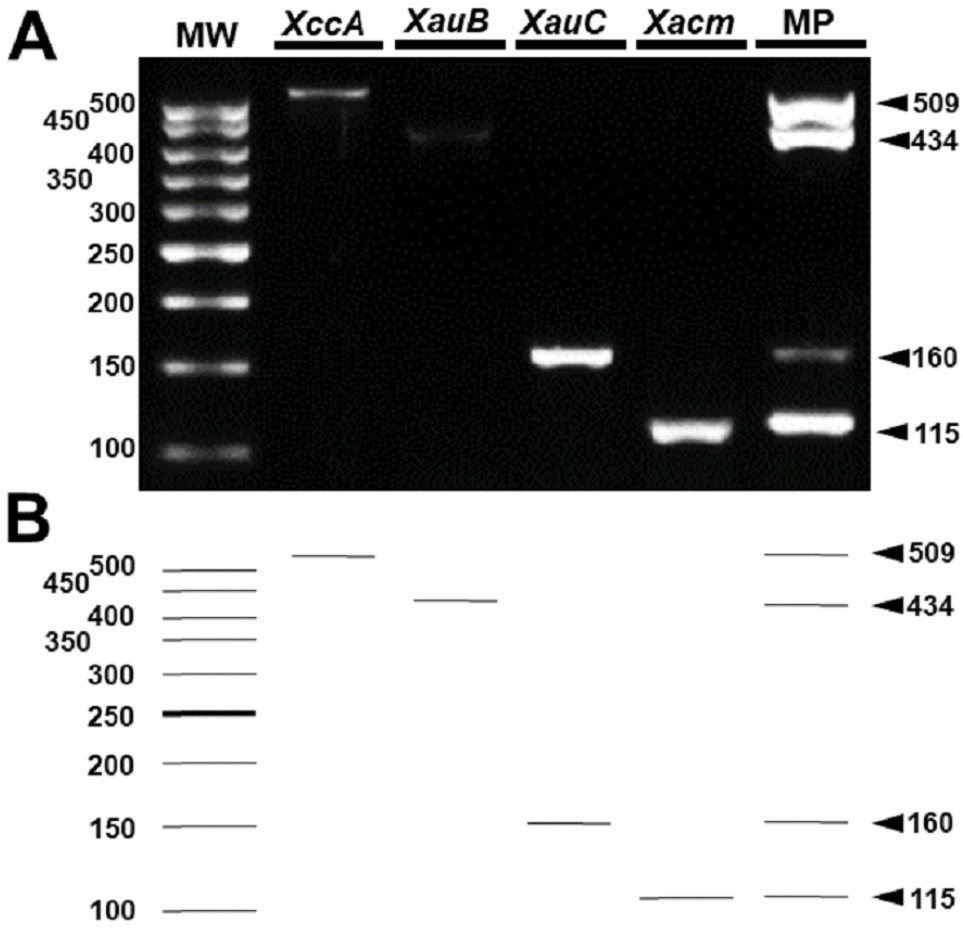

**Figure 4 Multiplex PCR of *Xanthomonas* that infect citrus.** (A) Multiplex PCR electrophoresis containing the combination of the DNA of one species with all pairs of specific primers generating only their amplicon (lines 2–5). MP—Combination of DNAs of all pathotypes with all primers (line 6). MW, molecular weight (ladder 100 bp) and (B) Theoretical electrophoresis of multiplex PCR.

the pathogen is crucial to restrain this disease, given that the management of plants during cultivation and export may differ with each infecting strain (*Graham et al., 2004*).

Due to its high virulence and host range, the development of diagnostics is aimed towards the detection of *XccA* strains, the causative agent of Asian citrus canker. Currently, field detection is performed through rapid tests based on the immune-identification of its proteins as Xcc ImmunoStrip® Test, Agdia, Inc. (Catalog number: STX 92200), but there are several studies that have developed serological diagnoses for *XccA* (*Goto, Takahashi & Messina, 1980*; *Trindade et al., 2007*; *Afonso et al., 2013*; *Alvarez et al., 1991*; *Bach & Alba, 1993*; *Bach & Guzzo, 1994*; *Yano et al., 1979*). However, these tests have low sensitivity, and in some cases, false negative results have already been reported in phenotypically different strains of *XccA* (*Vernière et al., 1998*; *Gottwald et al., 2009*). Additionally, its high cost per sample is a significant disadvantage when compared to methods based on molecular biology techniques.

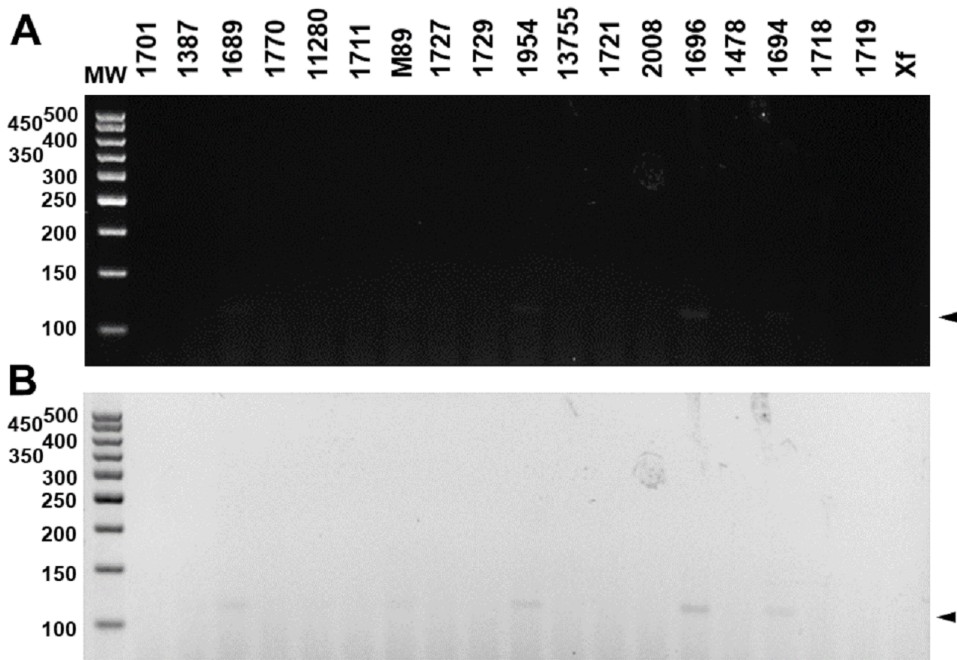

**Figure 5** **Multiplex PCR electrophoresis of *Xanthomonas* sp. that do not infect citrus.** (1701)—*X. axonopodis* pv. *vignicola* strain 1701; (1387)—*X. axonopodis* pv. *vesicatoria* strain IBSBF1387; (1689)—*X. axonopodis* pv. *vesicatoria* strain 1689; (1770)—*X. axonopodis* pv. *allii* strain IBSBF1770; (11280)—*X. axonopodis* pv. *phaseoli* strain IAC11280; (1711)—*X. alfafae* subsp. *alfalfae* strain 1711; (M89)—*X. axonopodis* pv. *passiflorae* strain M89; (1727)—*X. axonopodis* pv. *patelli* strain 1727; (1729)—*X. axonopodis* pv. *coracanae* strain strain 1729; (1954)—*X. axonopodis* pv. *manihotis* strain IBSBF 1954; (13755)—*X. axonopodis* pv. *phaseoli* var. *fuscans* strain IAC13755; (1721)—*X. citri* pv. *rhynchosiae* strain 1721; (2008)—*X. axonopodis* pv. *vasculorum* strain IBSBF 2008; (1696)—*X. axonopodis* pv. *axonopodis* strain 1696; (1478)—*X. axonopodis* pv. *dieffenbachiae* strain IBSBF 1478; (1694)—*X. axonopodis* pv. *glycines* strain 1694; (1718)—*X. citri* pv. *cajani* strain 1718; (1719)—*X. citri* pv. *sesbaniae* strain 1719; (Xf) *Xylella fastidiosa* strain 9a5c. MW, molecular weight (ladder 100 bp).

Some studies have already demonstrated the potential of molecular diagnosis for *Xanthomonas*, more specifically for *XccA* strains, based on several techniques such as conventional PCR, multiplex PCR, real-time PCR (qPCR), nested-PCR, loop-mediated isothermal amplification (LAMP-PCR), droplet digital polymerase chain reaction (ddPCR), BOX-PCR, and FISH (*Coletta-Filho et al., 2006*; *Mavrodieva, Levy & Gabriel, 2004*; *Munhoz et al., 2011*; *Ngoc et al., 2009*; *Park et al., 2006*; *Rigano et al., 2010*; *Waite et al., 2016*; *Zhao et al., 2016*; *Golmohammadi et al., 2007*; *Hartung, Daniel & Pruvost, 1993*; *Kositcharoenkul, Chatchawankanphanich & Bhunchoth, 2011*; *Cubero & Graham, 2001*; *Cubero & Graham, 2002*; *Cubero & Graham, 2005*). The wide variety of techniques used in the molecular diagnostics developed for citrus canker solved the problem of cost and low sensitivity presented by serological methods. However, the false-positive amplifications already highlighted among strains of genetically close variants, such as *XccA* and *Xau*, once again reveal the need for new detection methods that are specific and that distinguish the different pathotypes of these plant pathogens (*Delcourt et al., 2013*). Thus, to the best of our knowledge, this is the first study that presents a single method that simultaneously

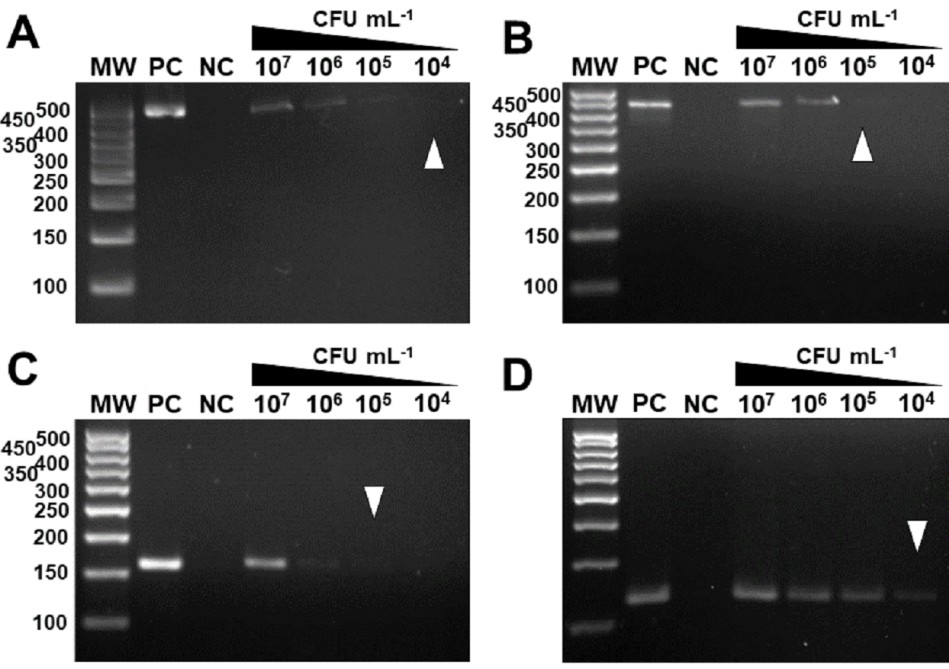

**Figure 6 Specificity and sensitivity of amplification by *in vivo* PCR.** Conventional PCR of bacterial exudate using the specific markers for *XccA*, *XauB*, *XauC*, and *Xacm*. The minimum concentration detected by PCR (A) for *XccA* was $1.5 \times 10^4$ CFU ml$^{-1}$, (B) for *XauB* was $1.5 \times 10^5$ CFU ml$^{-1}$, (C) for *XauC* was $1.5 \times 10^5$ CFU ml$^{-1}$, and (D) for *Xacm* $1.5 \times 10^4$ CFU ml$^{-1}$. PC, positive control. NC, negative control of leaves inoculated with water. MW, molecular weight (ladder 100 bp).

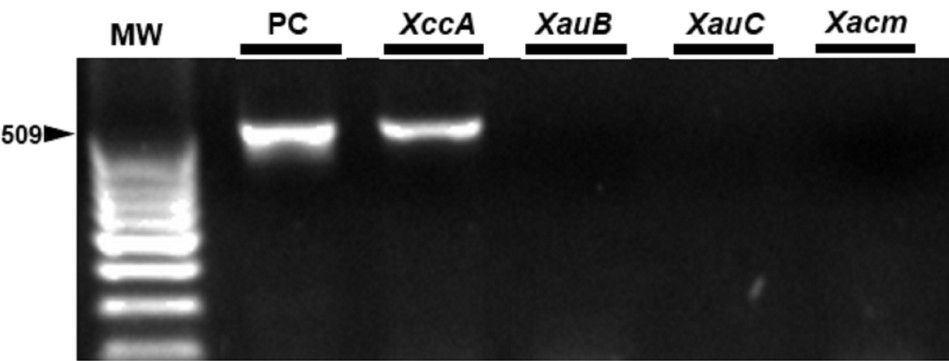

**Figure 7 Specificity of the markers in exudate samples from leaves inoculated by spraying with *XccA* strain.** PC, Positive control *in vitro* DNA extraction; *XccA*, *XauB*, *XauC*, and *Xacm* PCR using XccAm, XauBm, XauCm, and Xacmm specific markers respectively, for exudate samples of citrus canker lesions induced by *XacA* 306. MW, molecular weight (ladder 100 bp).

distinguishes the four pathotypes of *Xanthomonas* that infect citrus. In this method, each of these four pathotypes will have its own distinct amplicon.

It is important to emphasize that the method presented here is not a general method for identifying the *Xanthomonas* strains discussed here with respect to other bacterial isolates.

The application setting for which the detection tool was designed is for strains isolated from citrus trees only.

The specificity verified in the results enables the adoption of individualized management for each pathotype, which could reduce cultivation losses. Although BLASTn analysis of primer sequences against genomes deposited in NCBI has shown that non-specific binding to other *Xanthomonas* genomes may occur, two factors must be considered with respect to false positives: (a) a possible PCR product generated for other species of *Xanthomonas* would likely have a different size from that observed for citrus pathotypes; (b) as far as we know, the only members of the *Xanthomonas* genus that infect citrus are the ones considered here. Therefore we think that the probability of amplification of false positives in samples from diseased trees is small, which is corroborated by the analyses presented in Fig. 5.

The sensitivity of the PCR *in vivo* tests shows the potential of its usage as a preventive detection tool since it can detect as little as $10^4$ CFU of bacteria, allowing the pathogen to be screened during the colonization and survival phase of its life cycle, avoiding its later dissemination and infection of healthy plants. Remaining canker lesions under favorable conditions have a concentration of up to $10^7$–$10^8$ CFU of viable bacteria for a new infection (*Graham, Gerberich & Davis, 2016*; *Pruvost et al., 2002*).

Another potential benefit of the correct detection of *Xanthomonas* strains presented here is its use on seedlings that will be distributed to rural producers, since the introduction of infected seedlings in orchards is one of the mechanisms of dissemination of the bacteria to healthy plants (*Schubert et al., 2001*; *Schoulties et al., 1987*; *Das, 2003*). Additionally, our method can be used for a biogeographic survey of the pathotypes, to trace the distribution profile among citrus orchards throughout the world. This becomes important for epidemiological surveillance of this phytopathogen, as this type of analysis is typically hampered by the lack of adequate molecular typing techniques for surveillance and outbreak investigation (*Ngoc et al., 2009*).

Our approach is based on conventional PCR because it is an accessible and inexpensive technique, without compromising the quality of the analysis. Furthermore, the present study has established the detection of all pathotypes in the same PCR reaction, as demonstrated in the results of multiplex PCR. This technique increases the importance of the detection method, since the same sample can be submitted to a PCR reaction with all the primers, reducing the time for obtaining the result. Another technique used in our study, which could enhance the speed of detection, was the extraction of total DNA from the exudate of the leaves. This extraction takes less than four hours to be carried out; thus, there is no need to wait for several days for bacterial growth from its lesion isolation.

Collectively, both experimental procedures make the methodology feasible on a large scale and at a reduced cost. It is important to highlight that the variable size of the PCR products for each pathotype was established to facilitate its identification, and the use of PCR-RFLP would only be justifiable to verify genetic diversity within species, which is not the objective of this study. Moreover, although we did not use the TaqMan system to verify the potential of these molecular markers in the identification of these pathogens, it is likely that the use of this system would only make detection sensitivity higher than

that observed by conventional PCR, as described for other organisms (*Ranjbar et al., 2014*; *Angelone-Alasaad et al., 2015*; *Zhou et al., 2017*). However, the cost of analysis would be higher, which could significantly burden large-scale analyses.

Thus, our results not only allow the correct identification of the four pathotypes of *Xanthomonas* that infect citrus, but they also point to a possible bioinformatics pipeline for the production of species-specific molecular markers for any set of genomes defined by the user. The high specificity and sensitivity obtained demonstrated the effectiveness of our bioinformatics analysis.

## CONCLUSIONS

In summary, the identification of unique sequences from different genomes, using comparative genomics, allowed the development of specific molecular markers. This is the first conventional or multiplex PCR-based method that discriminates four different citrus-associated *Xanthomonas* pathotypes. This study further describes an effective pipeline for the generation of species-specific molecular markers.

## ACKNOWLEDGEMENTS

Thanks to all members of the Laboratory of Biochemistry and Molecular Biology (LBBM, Federal University of Ouro Preto, UFOP) and the Laboratory of Biochemistry and Molecular Biology of Faculty of Agrarian and Veterinary Sciences UNESP, Jaboticabal campus for their support. Thanks, Dr. Fabrício Jaciani, FUNDECITRUS researcher, for the critical help in finalizing this paper.

### Funding

The following agencies supported this work: the National Council of Technological and Scientific Development (CNPq Process 481226/2013-3), Foundation of Protection to Research of the State of Minas Gerais–FAPEMIG (process APQ-02387-14), Fundect-MS (007/2015 SIAFEM 025139) and Coordination for the Improvement of Higher Education Personnel (CAPES) (the BIGA Project, CFP 51/2013, process 3385/2013). Leandro Marcio Moreira, João Carlos Setubal, Nalvo Franco de Almeida, and Jesus Aparecido Ferro have a research fellowship from CNPq. The funders had no role in study design, data collection and analysis, decision to publish, or preparation of the manuscript.

### Grant Disclosures

The following grant information was disclosed by the authors:
National Council of Technological and Scientific Development: CNPq Process 481226/2013-3.
Foundation of Protection to Research of the State of Minas Gerais–FAPEMIG: process APQ-02387-14.
Fundect-MS: 007/2015 SIAFEM 025139.

Coordination for the Improvement of Higher Education Personnel (CAPES): BIGA Project, CFP 51/2013, process 3385/2013.

## Competing Interests

João C. Setubal is an Academic Editor for PeerJ.

Commercial use of the oligonucleotides and amplicons generated by PCR in this work is protected by Brazillian patent application number BR 10 2018 067332 7.

## Author Contributions

- Natasha P. Fonseca conceived and designed the experiments, performed the experiments, analyzed the data, prepared figures and/or tables, authored or reviewed drafts of the paper, approved the final draft.
- Érica B. Felestrino, Washington L. Caneschi, Angélica B. Sanchez and Deiviston S. Aguena performed the experiments, analyzed the data, prepared figures and/or tables, authored or reviewed drafts of the paper.
- Isabella F. Cordeiro, Camila G.C. Lemes and Renata A.B. Assis performed the experiments, authored or reviewed drafts of the paper.
- Flávia M.S. Carvalho performed the experiments, contributed reagents/materials/analysis tools, authored or reviewed drafts of the paper.
- Jesus A. Ferro conceived and designed the experiments, analyzed the data, contributed reagents/materials/analysis tools, authored or reviewed drafts of the paper.
- Alessandro M. Varani and Joao C. Setubal conceived and designed the experiments, analyzed the data, contributed reagents/materials/analysis tools, prepared figures and/or tables, authored or reviewed drafts of the paper, approved the final draft, revised the language.
- José Belasque and Guilherme P. Telles conceived and designed the experiments, contributed reagents/materials/analysis tools, authored or reviewed drafts of the paper.
- Nalvo F. Almeida and Leandro M. Moreira conceived and designed the experiments, analyzed the data, contributed reagents/materials/analysis tools, prepared figures and/or tables, authored or reviewed drafts of the paper, approved the final draft.

## Patent Disclosures

The following patent dependencies were disclosed by the authors:

The oligonucleotides and amplicons generated by PCR are protected by the patent application registration under process number BR 10 2018 067332 7. Updates on the patent status will be available at https://gru.inpi.gov.br/pePI/jsp/patentes/PatenteSearchBasico.jsp.

## Data Availability

The sequences are available on NCBI: NC_003919.1 (XccA306), CP011250.1 (XauB1561), CP011160.1 (XauC1559), and GCF_005059785.1 (Xacm1510).

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
