# Peer review of "Detection and identification of Xanthomonas pathotypes associated with citrus diseases using comparative genomics and multiplex PCR"

_PeerJ, doi:10.7717/peerj.7676_

## Round 0.1 · original submission · Minor Revisions

The reviewers recommended that you make minor amendments to your manuscript. Please respond to all comments raised by the reviewers and submit a revised version of your manuscript. We encourage you to submit your documents with tracked changes to highlight the revisions.

·

Basic reporting

This text of this manuscript is generally in clear, unambiguous and professional English. However, there are several exceptions where the meaning is unclear or ambiguous. There are a few minor lapses of grammar.

(1) In the Abstract, the authors first mention "three species" and then subsequently mention "four species". This is confusing for the reader. The confusion arises because there are three distinct pathogens and these collectively fall within three taxonomic species. The authors should carefully re-word the text to make this clear. See also point number (21) below.

(2) The authors write: "Although these species exhibit different levels of
virulence and aggressiveness, only limited alternatives are currently available for proper
and early detection of these diseases in the fields." I don't really follow the logic here. Why would the exhibition of different levels of virulence and aggression mean that there would be not be limited alternatives? In other words, (a) why use the word "although"? and (b) surely the alternatives will inevitably be limited, even if there are many. Please re-word this to make the intended meaning more clear.

(3) In the Abstract, the authors write: " It is freely available and can be utilized for any other model organism." Why specifically 'model' organism? I imagine that the most frequent scenario will be that a researcher wants to apply detection to non-model organisms rather than model organisms. For a defintion of 'model organism', see for example: https://doi.org/10.1016/j.endeavour.2013.06.001

(4) Line 62. "Gram-negative", not "gram-negative". This is named after Hans Christian Gram.

(5) The authors write "Xanthomonas is a genus of phytopathogenic gram-negative bacteria that cause a variety of diseases in several Citrus trees [1]." This is potentially misleading, since only a small minority of Xanthomonas bacteria cause disease in Citrus trees.

(6) Lines 65 and 66. The authors should not use italics for "sp." as it is not part of the taxonomic name. Furthermore, the authors need to consider whether they really mean "sp." (singular) or "spp." (plural) when they write "... for instance, X. oryzae in Oryza sp. (rice), X. campestris in Brassica sp. 66 (cabbage), X. vesicatoria in Solanum sp. (potato), and X. citri in Citrus sp. (orange) [2–6].". See also "Citrus sp." at lines 100- 101.

(7) Delete "that" from line 67.

(8) Line 70. Sometimes the authors write "Citrus" and sometimes "citrus". What is their rationale for distinguishing between these two terms? Would it be more helpful to the reader to be consistent?

(9) Line 73. I don't understand the expression "directly reflects on". What does this actually mean? Please use more precise language.

(10) Line 78. The authors write "Although they are genetically close to type A, these variants are related to a smaller set of compatible hosts (Table 1) [9,13].". It is confusing to say "related to" here because (a) the first part of the sentence concerns genetic relatedness, and (b) I don't really understand what related means here; does it mean something like 'associated with'? Or causal?

(11) Line 89. The authors write "... bacteria are classified among the main phytosanitary problems ...". this is confusing. The choice of the word 'classify' suggests taxonomic classification classification; but this is not what the authors mean. Do they simply mean that these bacteria present one/some of the most important phytosanitary problems?

(12) The authors write (line 296): "there are no reports available in the literature of other species of Xanthomonas not associated with citrus surviving in contact with this host, which would massively reduce the possibility of amplification of false-positives". That statement is true but it is not useful and the problem arises from the use of the word 'species'. For example, X. fuscans subsp. fuscans could be considered to be a member of a species associated with Citrus because the species X. fuscans contains the pathogen X. fuscans subsp. aurantifolii. Please carefully re-word this section to make the meaning clear and precise.

(13) Line 110 and Table 2. When refering to genome sequences obtained from public databases, it would be much more useful to provide the GenBank or RefSeq accession numbers of the actual genome sequence rather than the BioSample accession number for the biological sample from which those sequences were derived.

(14) Lines 113 and 214. "MAUVE" versus "Mauve"; please be consistent.

(15) Line 122. "set ... were" should be "sets ... were".

(16) Line 132. "compared against all the species deposited in the NCBI". Species are no deposited in NCBI; sequence data from species are deposited in various sequence databases available via the NCBI (and indeed other portals such as that of the EBI).

(17) What is meant by a "citrus dependent Xanthomonas isolate" (line 143)? In what sense are they 'dependent'? Perhaps the authors mean "Citrus-pathogenic"?

(18) The term "Electronic PCR" is confusing (line 208). Perhaps the authors really mean "in-silico PCR"?

(19) Line 228. "However, it does not occur upon using broad-coverage primers". What does not occur? Is it amplification that does not occur? Currently, the way it is written suggests that it is Stenotrophomonas which does not occur. Also, what is meant exactly by "broad-coverage primers"? Which primers, specifically, does this refer to?

(20) Line 261. "citrus canker is not the central pathology that currently affects citrus". What is meant by "central"? Does it mean something like "most prevalent"? Or "most economically important"? Or something else?

(21) Line 245. The authors write "In contrast, Xacmm amplified a weak fragment for five of the investigated species (X. axonopodis pv. vesicatoria 1689, X. axonopodis pv. passiflorae M89, X. axonopodis pv. manihotis IBSBF1954, X. axonopodis pv. axonopodis 1696, and X. axonopodis pv. glycines 1694) ...". They say "five ... species" but actually this is only one species, i.e. X. axonopodis.

Actually, it more complicated still, since X. axonopodis pv. manihotis has now been transferred into X. phaseoli (https://doi.org/10.1111/ppa.12461)!

(22) Line 305. "the diagnosis". I am not entirely sure that "diagnosis" is the correct word here. The purpose of the tool is detection, or perhaps identification, rather than diagnosis.

(23) There are some problems with formatting of references. See for example inappropriate use of capital letters in author names in reference number 6.

(24) Figure 1. "Primer Design Tool" should have capital initial letters as in line number 123, where the authors write "using Primer Designing tool [33]." Actually, if we look at the title of reference number 33, it suggests that the tool is properly called "Primer-BLAST"? Please use the correct name in both the text and the figure and be consistent in use of capital letters.

Experimental design

(1) The authors propose PCR-based tools.They could also consider discussing the possibility of adapting their approach to isothermal amplification methods such as LAMP, which might be more convenient in the field. Or at least they could add a sentence or two that justifies their choice of approach.

(2) In the Abstract, the authors say "The bioinformatics pipeline developed in this study to design specific genomic regions is capable of generating specific primers. It is freely available and can be utilized for any other model organism.". From where can the reader download this pipeline? In what format or programming language is it implemented?

(3) The authors, sensibly, check their primer sequences against available genome sequences to assess selectivity. However, it is important that the primers do not make false positives against other bacteria in the (normal/healthy) Citrus microbiome. The authors should consider searching the sequences of available microbiome metagenomic data such as from this study: DOI: 10.1128/AEM.00210-17.

Validity of the findings

(1) In the Introduction, lines 96-98, the authors write "the use of comparative
97 genomics as a tool for molecular diagnosis is still at a nascent stage. This is even more evident in the case of bacteria of the genus Xanthomonas".

This is quite a bold claim and the authors do not cite any references or present any other evidence to support this. Arr the authors aware of the numerous examples of genomics-informed molecular detection and diagnostics in Xanthomonas? Here are nine examples that I am aware of; there may be more:

https://doi.org/10.1111/j.1365-3059.2012.02654.x

and

https://doi.org/10.1111/ppa.12289

and

1: Larrea-Sarmiento A, Dhakal U, Boluk G, Fatdal L, Alvarez A, Strayer-Scherer A,
Paret M, Jones J, Jenkins D, Arif M. Development of a genome-informed
loop-mediated isothermal amplification assay for rapid and specific detection of
Xanthomonas euvesicatoria. Sci Rep. 2018 Sep 24;8(1):14298. doi:
10.1038/s41598-018-32295-4. PubMed PMID: 30250161; PubMed Central PMCID:
PMC6155141.


2: Lang JM, Hamilton JP, Diaz MGQ, Van Sluys MA, Burgos MRG, Vera Cruz CM, Buell
CR, Tisserat NA, Leach JE. Genomics-Based Diagnostic Marker Development for
Xanthomonas oryzae pv. oryzae and X. oryzae pv. oryzicola. Plant Dis. 2010
Mar;94(3):311-319. doi: 10.1094/PDIS-94-3-0311. PubMed PMID: 30754246.


3: Palacio-Bielsa A, López-Soriano P, Bühlmann A, van Doorn J, Pham K, Cambra MA,
Berruete IM, Pothier JF, Duffy B, Olmos A, López MM. Evaluation of a real-time
PCR and a loop-mediated isothermal amplification for detection of Xanthomonas
arboricola pv. pruni in plant tissue samples. J Microbiol Methods. 2015
May;112:36-9. doi: 10.1016/j.mimet.2015.03.005. Epub 2015 Mar 11. PubMed PMID:
25769438.


4: Wang H, Turechek WW. A Loop-Mediated Isothermal Amplification Assay and Sample
Preparation Procedure for Sensitive Detection of Xanthomonas fragariae in
Strawberry. PLoS One. 2016 Jan 14;11(1):e0147122. doi:
10.1371/journal.pone.0147122. eCollection 2016. PubMed PMID: 26766068; PubMed
Central PMCID: PMC4713083.


5: Wang XQ, Allen TW, Wang H, Peterson DG, Nichols RL, Liu A, Li XD, Deng P, Jia
D, Lu SE. Development of a qPCR Protocol to Detect the Cotton Bacterial Blight
Pathogen, Xanthomonas citri pv. malvacearum, from Cotton Leaves and Seeds. Plant
Dis. 2019 Mar;103(3):422-429. doi: 10.1094/PDIS-07-18-1150-RE. Epub 2019 Jan 11.
PubMed PMID: 30632895.


6: Nakato GV, Wicker E, Coutinho TA, Mahuku G, Studholme DJ. A highly specific
tool for identification of Xanthomonas vasicola pv. musacearum based on five
Xvm-specific coding sequences. Heliyon. 2018 Dec 27;4(12):e01080. doi:
10.1016/j.heliyon.2018.e01080. eCollection 2018 Dec. PubMed PMID: 30603713;
PubMed Central PMCID: PMC6307341.


7: Hodgetts J, Hall J, Karamura G, Grant M, Studholme DJ, Boonham N, Karamura E,
Smith JJ. Rapid, specific, simple, in-field detection of Xanthomonas campestris
pathovar musacearum by loop-mediated isothermal amplification. J Appl Microbiol.
2015 Dec;119(6):1651-8. doi: 10.1111/jam.12959. PubMed PMID: 26425811.


(2) The authors write at line 101 "These sequences are intended to be used
only for detecting isolates derived from Citrus sp.; these sequences do not distinguish citrus-associated Xanthomonas from any other bacteria."

I find this slightly puzzling. In order for the tool to successfully detect isolates from Citrus spp., it must be able to distinguish these from other bacteria. If it cannot distinguish the target organism from other bacteria, then there will be false positive results. For example, the microbiome of the Citrus tree must surely contain "other bacteria" and thus are a source of potential false positives. Furthermore, I don't really understand why the authors are being so cautious in this claim, given that they have experimentally demonstrated good specificity of their primers in silico, in vitro and in vivo?

I appreciate that in line 289, the authors write "It is important to emphasize that the method presented here is not a general method for identifying the Xanthomonas strains discussed here with respect to other bacterial isolates. The application setting for which the diagnostic tool was designed is for strains isolated from citrus trees only."

(3) Line 115. The "custom Perl scripts" must be made available, e.g. as supplementary information or submitted to a repository such as GitHub. See PeerJ Data and Materials Sharing policy.

(4) Line 214. "Mauve found a total of 5,217 genomic islands.". What is the definition of a 'genomic island'? One definition is "Genomic islands are clusters of genes within a bacterial genome that appear to have been acquired by horizontal gene transfer." (J.P. Ramsay, ... C.W. Ronson, in Reference Module in Life Sciences, 2017). Does Mauve actually check for evidence of horizontal transfer? Or does it simply look for sequences that are private to one genome or set of genomes?

(5) The authors present a new pipeline for detecting pathogen-specific sequence signatures. It would be beneficial to the reader if the authors could include some discussion about how their pipeline compares with previous approaches (why re-invent the wheel?) to solving this problem for example, but not limited to, these two previous publications:

1: Phillippy AM, Ayanbule K, Edwards NJ, Salzberg SL. Insignia: a DNA signature
search web server for diagnostic assay development. Nucleic Acids Res. 2009
Jul;37(Web Server issue):W229-34. doi: 10.1093/nar/gkp286. Epub 2009 May 5.
PubMed PMID: 19417071; PubMed Central PMCID: PMC2703920.


2: Phillippy AM, Mason JA, Ayanbule K, Sommer DD, Taviani E, Huq A, Colwell RR,
Knight IT, Salzberg SL. Comprehensive DNA signature discovery and validation.
PLoS Comput Biol. 2007 May;3(5):e98. Epub 2007 Apr 20. PubMed PMID: 17511514;
PubMed Central PMCID: PMC1868776.

Additional comments

This paper presents a method or pipeline for the design of pathogen-specific PCR primers and successfully applies it to the case of distinguishing among several important Xanthomonas pathogens of Citrus crops. It makes a useful contribution and is essentially scientifically sound, notwithstanding the specific issues that I raised above.

Reviewer 2 ·

Basic reporting

Overall, the paper is well presented and researched. The authors have conducted a very through literature review and provided good context.

Suggest that the introduction provides more detail or makes it clearer on how their work is novel compared to previous work on citrus canker diagnostics. For example, is this the first genome informed analysis to design PCR primers for Xanthomonas pathogens on citrus, some of the current published PCR tests are over 14 years old (e.g. Cubero et al. 2005). Could consider describing some of the false-positive and test specificity issues with the current diagnostics e.g. Delcourt et al (2013) (cited reference 52).

The concluding sentences in the introduction (lines 100 -102) are a bit ambigous and needs to be rewritten to clarify what the developed assay targets. For e.g. on one hand the paper states that their target sequences are specific to bacteria on citrus but then say they do not distinguish citrus associated Xanthomonas from any other bacteria. From their results there assay is specific and targets the 4 Citrus pathotypes.

The paper has 7 figures that could be condensed. The figures only need to present results that are representative. For e.g. Figure 5 is a photo of a blank gel with negative results (should at least provide a positive control on the gel).

The discussion could be revised to make some of the sentences clear.

Experimental design

The experimental design is robust. The papers describes the key components of assay design e.g. primer design, specificity, and sensitivity.

Specificity - a good in silico evaluation was good conducted. Lab testing of the target strains were reasonably well covered. More lab testing of the expected background microflora on citrus plants would have been useful to further support their findings. For example, testing of environmental and saprophytic strains - Pantoea agglomerans, Enterobacer sp, Stenotrophomonas sp etc.

Line 123 - when you refer to Primer Designing tool - is this Primer-Blast on NCBI?\

Line 140 - receipe for nutrient agar - "meat extract" should be beef extract.

Line 208 - useful to have a sentence to describe what eletronic PCR does? Note: NCBI replaced electronic PCR with Primer BLAST.

Validity of the findings

The results for testing assay specificity and sensitivity are robust and well supported.

Would be useful to describe how you assessed repeatability and robustness of the assay. For example, can you use different reagents and DNA extraction methods and get the same result, are results consistent when you test other citrus varieties, how were samples replicated.

Do you get a competition effect amongst the different primers when using multiplex PCR on all 4 targets at the same time. Figure 6 - suggest to might as the amplicon at 160 bp is much fainter than the other targets.

The discussion suggests that TaqMan could be used or developed with these primers to enable better test sensitivity. TaqMan is only likely to work with the smaller amplicon targets (<300 bp).

No information is provided to what region of the genome the primers amplify. For e.g. are the amplicons derived from known genes or genetic regions, what do the amplicon sequences match when you blast them?

---

## Round 0.2 · accepted · Accept

I believe the comments of the reviewers have been addressed correctly.